# OpenReview forum: "Provable Length Generalization in Sequence Prediction via Spectral Filtering"
_ICML.cc/2025/Conference — ICML 2025 poster_

### Official Review · Reviewer_Cycx · 2025-03-10

**Overall Recommendation:** 3

**Summary:**

- This paper considers the problem of length generalization for sequence prediction in linear dynamic systems.
- They define a new notion of regret, called Asymmetric Regret, which measures the difference in cumulative loss between an algorithm that is only allowed to use information in the past $L$ time points and the cumulative loss of a policy that is allowed to use information in the past $L'$ time points, where $L' \geq L$.
- The authors provide two spectral-filtering based algorithms, based on what assumptions one places on the LDS, that obtain sub-linear asymmetric regret when one takes $L' = T$ and $L = T^q$ for $q \in [0, 1]$.  The first algorithm obtains sublinear regret for any $q \in [0, 1]$, as long as the LDS satisfies certain constraints on its eigenvalues. The second algorithm removes the assumption needed on the LDS, but requires that $q \geq \frac{1}{4}$.
- Finally, the authors corroborate their theoretical findings with experiments.

## Update after rebuttal
I thank the authors for their response. As they have addressed most of my questions and concerns, I will maintain my positive score for this paper.

**Claims And Evidence:**

Yes, the claims made in the submission are supported by clear and convincing evidence.

**Essential References Not Discussed:**

No, I believe that the authors adequately summarized existing related works.

**Experimental Designs Or Analyses:**

Yes, I reviewed the experiments in Section 4, namely the experiments in Section 4.1.

**Methods And Evaluation Criteria:**

Yes, the proposed evaluation criteria, namely Asymmetric regret, makes sense for the problem at hand.

**Other Comments Or Suggestions:**

- Typo in equation (1)? Should is be $y_{t} = C x_t + D u_t + \zeta_t$? Likewise, in the equation in the right hand side column around likes 41, shouldn't $u_t$ be used when making a prediction about $\hat{y}_t$?

**Other Strengths And Weaknesses:**

**Strengths**:
- The paper is well-written and easy to follow
- I found the notion of Asymmetric regret to be interesting and this work to be timely due to the recent interest in length generalization for LMs

**Weaknesses**:
-  The connection to LLMs is unclear. In lines 40 - 44, the authors make it a point to emphasize the difficulty of length generalization in the context of LLMs. However, to me, it's unclear how studying length generalization for LDS has any bearing to length generalization for LLMs. It would be great if the authors can discuss, at a high-level, what implications the main take aways from this paper have on length generalization in LLMs. I think this is important, because it's not clear to me why one should care about length generalization for LDS (i.e why would I run Algorithms 5 and 6 when learning LDS?)

- Limited Empirical Evaluation. The authors effectively only run a single experiment. It would be nice to see the empirical performance of Algorithms 5 and 6 across a larger set of experiments, perhaps using real-world data. Moreover, the authors provide two algorithms, but don't provide an empirical comparison between them. I would be interested in seeing how the performance of Algorithm 5 and 6 compare, and whether the requirement that $q \geq \frac{1}{4}$ for Algorithm 6 is really needed in practice.

**Questions For Authors:**

See weaknesses above.

**Relation To Broader Scientific Literature:**

Length generalization, the ability for learning algorithms trained using one context window size to generalize to much longer context windows, has even extensively studied empirically in the context of large language models. This paper initiates the theoretical study of length generalization in the context of learning linear dynamic systems. To the best of my knowledge, this is the first paper to formalize this problem in the context of linear dynamical systems.

**Theoretical Claims:**

I did not check the correctness of the proofs of the Theorem statements.

---

> ### Author Rebuttal · Authors · 2025-04-01
>
> Thank you for your thoughtful and detailed review!
>
> Connection to LLMs Response
>
> You are correct - we have zero theory for LLMs, and unfortunately this is not uncommon, our theoretical understanding of LLM is very limited. We see this as a start of a theory for length generalization for *any* sequence, and the easiest sequence we could start with is that for linear dynamical systems. It is rich enough to have been used in LLM applications, please see papers [2,3,4], and yet amenable to analysis. In section 4.2 we show that despite our analysis only holds for the basic mathematical model of linear dynamical systems, our techniques actually do apply for more sophisticated signals, showing evidence that the theory can be enhanced in the future to incorporate the language signal.
>
>
>
> Empirical Validation Response
>
> Our experiments are proof-of-concept, and are not immediately applicable to LLMs, although we certainly hope they will in the future. We assume you mean comparing the performance of Algorithm 1 and 2 (not 5 and 6?). In fact, we do compare them on page 7, showing that Algorithm 2 is much more robust to the spectrum of $A$. That's a great suggestion to include plots for $L = T^{1/4}$ and smaller in Figure 4. We will do this.
>
>
>
> [2] Resurrecting Recurrent Neural Networks for Long Sequences
> Antonio Orvieto, Samuel L Smith, Albert Gu, Anushan Fernando, Caglar Gulcehre, Razvan Pascanu, Soham De
>
> [3] Efficiently modeling long sequences with structured state spaces
> A Gu, K Goel, C Ré
>
> [4] Spectral state space models
> N Agarwal, D Suo, X Chen, E Hazan

---

> > ### Comment · Reviewer_Cycx · 2025-04-04
> >
> > I thank the author for their response and for addressing my concerns. I will maintain my positive score.

---

### Official Review · Reviewer_usFi · 2025-03-13

**Overall Recommendation:** 4

**Summary:**

The paper introduces a novel theoretical framework addressing length generalization in sequence prediction tasks using spectral filtering methods. It defines a new metric, Asymmetric-Regret, that quantifies the regret of predictors trained with shorter contexts against those trained with longer ones. The authors provide rigorous proofs demonstrating that spectral filtering predictors can achieve provable length generalization under certain conditions on the linear dynamical system (LDS). A gradient-based learning algorithm is proposed that provably achieves length generalization for linear dynamical systems. Experimental validations on synthetic datasets confirm theoretical predictions.

**Claims And Evidence:**

The paper makes clear theoretical claims supported by rigorous proofs. However, experimental evidence is limited to synthetic data, focusing on linear dynamical systems. Broader applicability to nonlinear or noisy settings is not fully explored.

**Essential References Not Discussed:**

Not to my knowledge

**Experimental Designs Or Analyses:**

The synthetic experiments are well-designed to validate theoretical predictions. They clearly demonstrate the conditions under which length generalization occurs.

**Methods And Evaluation Criteria:**

The proposed spectral filtering methods are theoretically well-justified and the evaluation via Asymmetric-Regret is suitable for the studied context.

**Other Comments Or Suggestions:**

1. Incorporate bibliography into the main manuscript rather than supplementary material.
2. Minor: Use standard notation for real numbers (\mathbb{R} instead of \mathcal{R}).
3. Minor: Juxtapose Algorithms 1 and 2 for easier comparative readability.

**Other Strengths And Weaknesses:**

### Strengths
1. Clearly articulated and rigorous theoretical analysis.
2. Introduction of the insightful Asymmetric-Regret metric.

### Weaknesses
1. The notion and details of Spectral Transform Unit (STU) need further clarification and motivation.

**Questions For Authors:**

1. Could you provide examples of realistic systems where the spectral assumptions are met?

2. In Induction Heads Task, why assume the rest of the sequence consists of the same special blank token? Is this simplification necessary for theoretical clarity or could it be relaxed?

3. In Figure 5, why does the upper boundary of confidence intervals exceed accuracy = 1? Clarify missing footnote 3.

**Relation To Broader Scientific Literature:**

This paper clearly follows a series of works by Hazan et al. (2016, 2017a, 2017b, 2018, 2020). The introduction of Asymmetric-Regret and the theoretical guarantees provided for spectral filtering represent a meaningful extension of this line of work.

**Theoretical Claims:**

The correctness of the theoretical proofs (Theorems 1, 2, 4, 5, 6) appears solid within the assumptions stated.

---

> ### Author Rebuttal · Authors · 2025-04-01
>
> Thank you for a thoughtful and detailed review!
>
> Q1 (Realistic Systems)
>
>  This is a good question. On the face of it - linear dynamical systems are a toy mathematical model, and it is unclear if real dynamics are linear or symmetric. However: 1) our bounds do not depend on the hidden dimension, so one can imagine very complex dynamics that we only see their projection. 2) Some previous papers such as [2,3,4], argue that LDS have good modeling capacity for language, even with symmetric transition matrices. 3) Our work extends to very recent advances in spectral filtering by [5], such that it extends to more general linear dynamical systems.
>
> Q2 (Induction Heads Task)
>
> This is only for simplifying the setting, there is no need for this assumption, and we could have experimented with random extensions just as well.
>
> Q3 (Figure 5)
>
> This is just a notation, you are 100\% correct that above 1 has no meaning, but the standard mathematical notation is mean+std, which can be larger than 1, so we went with that. Our preference would be to cut off at 1.  Missing footnote 3 said "Even though the accuracy cannot go above $1$, the error bars are still well defined above this value." We will make sure this compiles correctly.
>
>
> References
>
>
> [2] Resurrecting Recurrent Neural Networks for Long Sequences
> Antonio Orvieto, Samuel L Smith, Albert Gu, Anushan Fernando, Caglar Gulcehre, Razvan Pascanu, Soham De
>
> [3] Efficiently modeling long sequences with structured state spaces
> A Gu, K Goel, C Ré
>
> [4] Spectral state space models
> N Agarwal, D Suo, X Chen, E Hazan
>
> [5] Dimension-free Regret for Learning Asymmetric Linear Dynamical Systems
> Annie Marsden, Elad Hazan

---

> > ### Comment · Reviewer_usFi · 2025-04-08
> >
> > I thank the authors for their replies. My concerns and questions are resolved and I will keep my rating.

---

### Official Review · Reviewer_9Uxb · 2025-03-14

**Overall Recommendation:** 4

**Summary:**

The authors consider the problem of length generalization in sequence prediction, i.e., whether online time series prediction algorithms can learn long-range dependencies using only a short context window during training. Despite its importance in areas like LLMs, the current literature offers few theoretical analyses and guarantees.

The authors introduce a new regret metric that compares the performance of a short-context learner against the best predictor using a longer context. They highlight spectral filtering algorithms as strong candidates for theoretically-provable length generalization and adapt them for short-context settings. In this setting, they derive theoretical guarantees for Asymmetric Regret in LDS under specific spectral conditions on the system matrix A. Additionally, they introduce an alternative spectral filtering method which uses two autoregressive components and has similar regret bounds but with much weaker constraints on A. The validity and sharpness of these bounds are demonstrated through sequence prediction experiments. Finally, they provide a proof-of-concept experiment on length generalization in nonlinear tasks, using a spectral filtering-based deep learning architecture.

**Claims And Evidence:**

The theoretical results in this paper are presented in an understandable and convincing way, and they are clearly supported by Figures 2, 3, and 4.

**Essential References Not Discussed:**

The literature is sufficiently covered.

**Experimental Designs Or Analyses:**

The experiments supporting the theoretical results provided by Theorems 5 and 6 are well-executed and sound.

I am not entirely convinced of the significance of Section 4.2, aside from vaguely motivating a future work direction about length generalization via STUs. The section does not directly use any of the theoretical results derived earlier in the paper and, as far as I understand, demonstrates that a deep learning architecture based on spectral filtering exhibits some degree of length generalization. It is unclear to me how one should interpret this result, as it is not given in comparison to other deep learning architectures (RNNs, Transformers, SSMs..), which presumably also have some generalization properties.

**Methods And Evaluation Criteria:**

The methods and evaluation criteria make sense for the problems discussed, though I wonder if Figure 5 should be zoomed out a bit more so the reader can see the whole confidence intervals.

**Other Comments Or Suggestions:**

I believe that the STU acronym is currently not defined in the text, and I think it would be useful to include a sentence in Section 4.2 explaining what STUs are.

**Other Strengths And Weaknesses:**

See my comments above.

**Questions For Authors:**

1) Your theorems assume a noiseless LDS. How robust are Algorithms 1 and 2 in settings with noise? Could they be applied directly to real-world datasets while still exhibiting long-range memorization?

2) What is the intended takeaway from Section 4.2, and do you think it would benefit from benchmarking against other models?

**Relation To Broader Scientific Literature:**

This paper is a clever and important contribution to the theory behind the long-range memory mechanisms of sequential models, which is subject to many empirical studies. The authors provide strong insights into the theoretical guarantees of length generalization of spectral filtering algorithms in the case of linear systems and suggest interesting future work directions for non-linear systems.

**Theoretical Claims:**

I briefly looked at the proofs in sections A, B, and C in the Appendix, and the reasoning made sense to me. I am not an expert on this topic, though.

---

> ### Author Rebuttal · Authors · 2025-04-01
>
> Thank you so much for your thoughtful and detailed review!
>
> Q1 (noisy setting)
>
> Our theorem extends to the case of stochastic noise with bounded variance. To extend to the case of adversarial noise, the norm of the allowed noise must be bounded with inverse to $T$. We present our results in the noiseless case for simplicity of the presentation and proof, but we can happily provide a corollary or note on this. Note that our theorem is presented in the online regret-minimization setting with respect to the best spectral filtering predictor in hindsight. This predictor gets $O(\sqrt{T})$ regret wrt the best linear dynamical predictor for *any* signal, regardless of the noise (shown in Hazan, E., Singh, K., and Zhang, C. Learning linear dynamical systems via spectral filtering. Advances in Neural Information Processing Systems, 30, 2017b).
>
>
> Q2 (Section 4.2)
>
> We think of 4.2 demonstrating that while our theory applies to LDS, it can be more widely applicable even to LLM tasks. It is a proof of concept, and we don't make a strong claim here, we just thought this will be useful for future scientists to explore. We are willing to remove it, the paper has a lot of results.  We are also willing to include benchmarking from other models.

---

### Official Review · Reviewer_BkYk · 2025-03-24

**Overall Recommendation:** 3

**Summary:**

The authors study an online sequence prediction problem where the sequence is generated by a time-invariant linear dynamical system within a class of spectral filtering predictors. They introduce a notion of regret between spectral filters that use context length $L$ and the full context, i.e., length $T$. Two algorithms are given to learn spectral filters with limited context and their regrets based on the best possible predictor with full context is shown sublinear, roughly of order $1/\sqrt{T}$. The first algorithm uses a single autoregressive term ($y_{t-1}$) and the learning guarantees require assumptions on the spectral properties of the ground truth linear dynamical system. The second algorithm incorporates an additional autoregressive term ($y_{t-2}$) with a different spectral filter to have robust estimation even without the spectral assumption.

### Update after rebuttal

I have increased my score following discussions with the authors, assuming they will follow through on their commitments to (i) provide a deeper discussion of length generalization in this work and the related literature, and (ii) highlight the notion of regret and the $L \ll T^{1/4}$ case, as described in their rebuttal.

**Claims And Evidence:**

I leave the discussion of the technical results to "Theoretical Claims". The experiments in Section 4.1. supports the main technical results on two algorithms presented and convincing. The claims in Section 4.2. on STU is not particularly detailed or convincing.

**Essential References Not Discussed:**

I don't think the related length generalization literature is discussed in depth in the paper. See [1] and references within:

[1] Anil, Cem, et al. "Exploring length generalization in large language models." Advances in Neural Information Processing Systems 35 (2022): 38546-38556.

**Experimental Designs Or Analyses:**

Experiments in Section 4.1. seem valid with a good design to check the main technical results. Experiments in Section 4.2. are unclear and I am skeptical towards the analysis provided by the authors. There are no details regarding how the experiments are conducted, including the supplementary material.

**Methods And Evaluation Criteria:**

The experiments in Section 4.1. is well-suited for the main technical claims of the paper. The experiments in Section 4.2. is to relate the findings to spectral transform units and makes sense to see how the theory extends to these architectures.

**Other Comments Or Suggestions:**

1. The definition of context-length for an online predictor needs to be clarified. The wording L167-170 is a bit misleading. I believe the form of the predictor is such that it depends only on the previous L timesteps but the choice of predictor is of course dependent on all the trajectory.
2. In algorithm 1, do you project to $\mathcal{K}_r$?

**Other Strengths And Weaknesses:**

1. The strength of the paper is that it proves a very curious statistical result. This technical contribution seems original and conveyed clearly.
2. The main weaknesses is that the paper is presented around the topic of length generalization but my judgment is that this is very misleading. Therefore, I suggest the authors to revise the manuscript, clarify the relationship to the length generalization literature (if there is any) and write a paper that focuses on their technical result.

**Questions For Authors:**

1. Could you explain my question regarding $\phi_i$ that is in “Theoretical Claims” section?
2. Could you please respond to the comment regarding relation to the broader scientific literature.

**Relation To Broader Scientific Literature:**

I believe the definition of length generalization used in this paper does not necessarily match the the notion of length generalization in the broader scientific literature, especially the literature on reasoning.

In this paper, the authors have a task in which they are able to statistically solve with a shorter context length than that of the full sequence. This is atypical in reasoning tasks such as addition, multiplication etc. where the part of the context only gives partial information for the final output. Therefore, in these settings, length generalization refers to the ability of models to apply a certain algorithm learned from shorter context windows to longer sequences. An example would be a model that has been trained in 3 digit summation that generalizes to 5 digit summation.

I believe this distinction needs to be clarified and presented better in the paper. It is unclear how much of the interest in length generalization relates to the setting authors has.

**Theoretical Claims:**

There seems to be a minor issue in the algorithms. $\phi_i$ are eigenvectors of $H_T$ which is of size $T x T$. Hence, $\phi_i$ has the size $T$ whereas it needs to be of size $L$. Could the authors clarify how to get spectral filters of size $L$? Is it just simple truncation? As far as I checked, this is not addressed in the appendix as well.

I have not checked the correctness of two theoretical claims.

---

> ### Author Rebuttal · Authors · 2025-04-01
>
> Thank you so much for your thoughtful and detailed review!
>
> Q1 (L vs T)
>
> In the paper we distinguish between L and T, where L is the context length and T is the overall sequence length. It is possible to take L=T. You are right that $\phi_i$ has size $T$ but we only allow ourselves to look at $L$ previous inputs and so there is indeed a dimension mismatch if $L < T$. The way we deal with this is by zero-padding the inputs. Specifically, the matrix which stacks $u_t, \dots, u_{t-L}$ would instead stack $u_t, \dots, u_{t-L}, 0, \dots, 0$. We state this on page 1 under ``Our Contributions" but we can make it a larger/more-clarified point.
>
>
> Q2 (Broader Scientific Literature)
>
> This is a great point, and we can give an answer, which we think will improve the paper. The reviewer observes that often in length generalization, the predictor is only allowed to learn/train on short context length, and then predicts on long context length and, importantly, in order for the prediction to be correct, the algorithm must make use of the full long context. In the online learning/regret setting the analogous setting would be to restrict the learning algorithm to context length $L<T$ when making gradient updates (ie keep the algorithm in our paper as is) but to measure its performance when making predictions on the full context length. We agree that we should include this notion of regret in our paper. Note that our theoretical results immediately apply to this setting, since the spectral filtering predictor is only given more power. Indeed, the regret bounds provided in our paper would hold for this notion of regret and, furthermore, they would be stronger-- they would hold without any assumption on the spectrum of $A$ with respect to $L$. Of course -- the above works only if the signal comes from a linear dynamical system with the properties assumed in the paper. We will also include a small discussion on which signals we believe would distinguish between these two notions of regret (i.e. when is our original notion of regret too pessimistic for an algorithm to succeed and when must we move to the notion where predictions use the full context?).
>
> Essential References Response
>
> We will cite [1] as a practically relevant paper, notice our treatment is theoretical and in a broader setting of sequence prediction for linear dynamical systems, not necessarily language.
>
>
> Main Weaknesses Response
>
> We suspect the reviewer will have a different opinion in light of our clarification of the above theorem and the fact that our results apply to that setting. However, we are happy to emphasize further that our paper is a theoretical one and considers length generalization for general sequence prediction in dynamical systems, rather than an empirical paper on methods that are useful for LLMs to length generalize.
>
> Other Comments
>
> 1. We completely agree and will change this phrasing.
> 2. Algorithm 1: projection can be applied to any convex set K, but the most common one (and the one we use) is with $K_r$ for some Euclidean diameter bound r.

---

> > ### Comment · Reviewer_BkYk · 2025-04-03
> >
> > Thank you for your responses.
> >
> > My main criticism concerns the nature of the task. The authors demonstrate that the task does not require the full context but only a short context. Referring to this as "length generalization" seems like a misuse of language. To clarify my position, I have no concerns about its usefulness for LLMs.
> >
> > Lastly, in response to Q2, I have updated my score. I believe this clarification is important. However, as noted above, the more significant clarification pertains to the statistical nature of the task and the usage of "length generalization".

---

> > > ### Author Response · Authors · 2025-04-04
> > >
> > > Thank you for reading our reply and addressing it!
> > > In our setting , there is a case where the full context is required, and learning to use it from a shorter one is possible. In the regime where L << T^{1/4}, the context-length constrained predictor no longer performs well and hence the task is difficult. In this setting, our proof technique immediately shows that, even if the learner is constrained to only use the L most recent history, if the predictor gets to look at the full context length then it can still predict well (i.e. this new notion of regret that switches context length for learner and predictor is still sqrt(T)). We suspect the reviewer may find this setting the most intriguing (i.e. L << T^{1/4}) and we are very happy to highlight it in the paper.  We are also happy to add a deep discussion in the related work on the various notions of length generalization used in empirical work and how it compares to this setting.

---

### Decision · Program_Chairs · 2025-05-01

**Decision:**

Accept (poster)

**Comment:**

The reviewers are in consensus that this is a strong contribution to ICML.

They should implement the following before publication:
_(i) provide a deeper discussion of length generalization in this work and the related literature, (BkYk)_
_(ii) highlight the notion of regret and the case, as described in their rebuttal. (BkYk)_
They have also made promises to implement a round of changes in their own comments, which this AC encourages them to follow through on.